# Adherence to a Mediterranean Diet Protects from Cognitive Decline in the Invecchiare in Chianti Study of Aging

**DOI:** 10.3390/nu10122007

**Published:** 2018-12-19

**Authors:** Toshiko Tanaka, Sameera A. Talegawkar, Yichen Jin, Marco Colpo, Luigi Ferrucci, Stephania Bandinelli

**Affiliations:** 1Longitudinal study section, Translation Gerontology Branch, National Institute on Aging, Baltimore, MD 21224, USA; FerrucciLu@grc.nia.nih.gov; 2Department of Exercise and Nutrition Sciences, Milken Institute School of Public Health, The George Washington University, Washington, DC 20052, USA; stalega1@email.gwu.edu (S.A.T.); yjin@email.gwu.edu (Y.J.); 3Geriatric Unit, Azienda Sanitaria Firenze (A.S.F.), Florence 50125, Italy; marco.colpo@hotmail.it (M.C.); stefania.bandinelli@asf.toscana.it (S.B.)

**Keywords:** Mediterranean diet, cognitive decline, longitudinal analysis

## Abstract

Following a Mediterranean diet high in plant-based foods and fish, low in meat and dairy foods, and with moderate alcohol intake has been shown to promote healthy aging. Therefore, we examined the association between a Mediterranean diet and trajectories of cognitive performance in the InCHIANTI study. Subjects (*N* = 832) were examined every 2–3 years up to 18 years with an average follow-up period of 10.1 years. Cognitive performance was assessed using the Mini Mental State Examination (MMSE) at every visit. Dietary habits were assessed using a validated food frequency questionnaire and adherence to Mediterranean diet was computed on a scale of 0-9 and categorized into three groups of low (≤3), medium (4–5), and high (≥6). Those in the highest adherence group (OR = 0.48, 95% CI: 0.29–0.79) and medium adherence group (OR = 0.64, 95% CI: 0.41–0.99) were less likely to experience cognitive decline. The annual average decline in MMSE scores was 0.4 units, for those in the high and medium adherence group this decline was attenuated by 0.34 units (*p* < 0.001) and 0.16 units (*p* = 0.03), respectively. Our findings suggest that adherence to a Mediterranean diet can have long-lasting protective effects on cognitive decline and may be an effective strategy for the prevent or delay dementia.

## 1. Introduction

Dementia is a growing public health concern that is characterized by cognitive impairment in multiple domains, consequently leading to a loss of independence. The incidence of dementia increases with advanced age, and it is estimated that the number of affected individuals will double every 20 years to an estimated 115.4 million by 2050 [1]. With such a worldwide impact, it is of the utmost importance to identify strategies to prevent or delay the onset of dementia, particularly because there are currently no effective pharmacological treatments to alleviate the symptoms after disease onset [2].

The development and progression of cognitive impairment is multifactorial, with older age being the greatest risk factor [3]. Research suggests that the processes that eventually leads to cognitive decline and dementia begins many years (even decades) before the onset of symptoms [4,5,6]. This suggests that preventative strategies are more likely to be successful if implemented early in the process. Therefore, identifying modifiable risk factors that can be incorporated into daily preventative strategies, such as lifestyle changes, might have a significant public health impact in curtailing the incidence of dementia. To this end, following a Mediterranean diet (MedDiet), a dietary pattern characterized by higher intake of plant based foods (vegetables, fruits, nuts, legumes, cereals) and fish, high in monosaturated fats (primarily from olive oil), but low in meats, dairy, and saturated fats with moderate intake of alcohol, has been shown to have beneficial effects on the risk of various age-related pathology including neurological diseases [7].

Several longitudinal studies have investigated the effects of MedDiet on incidences of dementia and cognitive decline with inconsistent results. Some show that greater adherence is associated with reduced risk of dementia and slower cognitive decline [8], while other show no effect [9,10]. While the totality of evidence supports the protective effect of adherence to MedDiet on dementia and cognitive function, there are still some inconsistencies in the reported associations [11]. Some of the inconsistencies may be due to differences in study design, including differences in follow-up time. Therefore, we examined the associations between adherence to MedDiet with prevalent dementia and cognitive impairments in the participants of the InCHIANTI study. We also tested the hypothesis that adherence to a MedDiet was protective against subsequent cognitive decline over an 18-year follow-up.

## 2. Materials and Methods

### 2.1. Study Population

The InCHIANTI study is a population-based epidemiological investigation aimed at evaluating factors that influence mobility in the older population living in the Chianti region in Tuscany, Italy. The details of the study have been previously reported [12]. A total of 1616 residents were selected from the population registry of Greve in Chianti (a rural area: 11,709 residents with 19.3% of the population greater than 65 years of age), and Bagno a Ripoli (Antella village near Florence; 4704 inhabitants, with 20.3% greater than 65 years of age). The participation rate was 90% (*n* = 1453), and the subjects ranged from 21 to 102 years of age. Overnight fasted blood samples were collected and used for genomic DNA extraction, and measurement of ApoE genotype. A baseline visit was conducted between 1998–2000 (visit 1), followed by four follow up periods between 2001–2003 (visit 2), 2004–2006 (visit 3), 2007–2009 (visit 4), and 2016–2017 (visit 5). For the cross-sectional analysis, we used data from 1139 subjects 65 years or older at baseline with cognitive data. For the longitudinal analysis, 832 subjects who were not cognitively impaired (definition described in Section 2.3) at baseline were included in the analysis. The study protocol was approved by the Italian National Institute of Research and Care of Aging Institutional Review and Medstar Research Institute (Baltimore, MD, USA).

### 2.2. Dietary Assessment and Construction of Mediterranean Diet Score

Usual dietary consumption in the past year was assessed at baseline using a food frequency questionnaire (FFQ) created for the European Prospective Investigation on Cancer and Nutrition study that was validated for the InCHIANTI study [13]. The Mediterranean diet score (MDS) was computed using daily dietary intakes derived from the FFQ following the algorithm developed by Trichopoulou et al. [14], as previously described [15]. The intakes of nine food groups were dichotomized using sex-specific median values as cutoff. A score of 1 was assigned for consumption equal to or above the median level of presumed beneficial foods (vegetables, legumes, fruits, cereal, fish, and ratio of monounsaturated fats:saturated fats (MUFA:SFA)) and for consumption equal to or below the median level of presumed detrimental foods (meat and dairy products) and a score of 0 for any other reported consumption. For alcohol, 1 point was assigned to men who consumed between 10 and 50 g/d and to women who consumed between 5 and 25 g/d versus a score of 0. Thus, the total MDS ranged from 0 (minimal adherence to the traditional MedDiet) to 9 (maximal adherence). For the analysis, the score was categorized into three groups as follows: low adherence (MDS ≤ 3), medium adherence (MDS 4–5), and high adherence (MDS ≥ 6).

### 2.3. Assessment of Cognitive Status at Baseline and Cognitive Decline

Assessment of participants’ cognitive status at baseline was conducted using a two-stage screening procedure, as previously described [16]. Global cognition was assessed using an Italian version of the Mini Mental State Examination (MMSE) [17]. In addition, participants who reported difficulty in performing activities of daily living (ADLs) or instrumental activities of daily living (IADLs) were asked whether the difficulty in performing these tasks was due to cognitive impairment. Those with a score MMSE score >26 were considered not to have dementia, while those with a score ≤21 were considered possibly having dementia and directly scheduled for the second stage screening procedure. The participants with a MMSE score between 22 and 26 received additional neuropsychological tests assessing memory (paired words test), concentration or attention (digit test from the Weschler adult intelligence test), and visuo–spatial ability (the Caltagirone drawings) [18], and MMSE was re-calculated based on education-adjusted normative score. The participants for whom the new score was >26 were considered “without dementia”, while those for whom the newly calculated score remained between 22 and 26 were scheduled for the second stage screening. The second stage screening was performed by geriatricians and a psychologist with long standing clinical experience in the evaluation of older patients with cognitive impairment. A diagnosis of “dementia syndrome” independent of the etiology was established using a standard evaluation protocol based on the DSM IV criteria. 

For this analysis the study population was categorized into three groups: (1) participants with normal cognitive functions (i.e., MMSE score ≥23, no diagnosis of dementia and no disability attributable to cognitive impairment); (2) participants with cognitive impairment but without dementia (i.e., those with a MMSE score <23 and/or any degree of disability in ADLs or IADLs that was judged to be related to impaired cognitive function) ; and (3) participants affected by a dementia syndrome established according to the criteria reported above. Change of cognitive function over time was assessed as a change in MMSE scores from baseline to follow-up of 1, 2, 3, 4, or 5, in participants with normal cognitive function at baseline. In the analyses, cognitive decline was defined as a 5-point decrease in MMSE score at any of the follow-up visits [19]. Trajectories of MMSE decline were calculated using linear mixed models as described in the statistical methods section.

### 2.4. Measurement of Main Covariates

Factors that were associated with cognitive function in univariate analyses or known to be associated with cognition from previous studies were considered as covariates in the analyses. Sociodemographic information included age, sex, and years of education, which was obtained during the structured interview. Smoking status was assessed by self-report, and participants were categorized into never smokers, former smokers or current smokers (smoking within 3 years of interview). The level of physical activity in the 12 months prior the interview was assessed through a modified standard interview-administered questionnaire and was coded into 3 categories of low activity (inactivity or light-intensity activity <1 h per week), medium activity (light-intensity activity 2–4 h per week), and high physical activity (light-intensity activity at least 5 h per week or moderate activity at least 1–2 h per week) [20]. Height and weight were measured in the clinic and body mass index (BMI) was calculated as weight in kilograms divided by height in meters squared. Inflammatory markers included C-reactive protein (CRP) measured using enzyme-linked immunosorbent assay (ELISA) and colorimetric competitive immunoassay (Roche Diagnostics, GmbH, Mannheim, Germany) and IL-6 measured using Bio-source cytoscreen ultra-sensitivity kits. Plasma fatty acids omega-3 and omega-6 were measured using gas chromatography, as previously described [21]. Circulating carotenoids and alpha tocopherols were measured via high-performance liquid chromatography (HPLC) [22,23]. Chronic diseases including hypertension, diabetes, ischemic heart disease, congestive heart failure, stroke, chronic obstructive pulmonary disease, cancer, Parkinson’s disease, hip fracture and lower extremities joint disease, anemia, kidney disease, and peripheral artery disease were defined using standard clinical definitions, which have been described previously [24]. Apolipoprotein E (ApoE) variant genotypes (e2, e3, e4) were defined by two single nucleotide polymorphisms (SNPs), rs429358 and rs7412. Genotyping of these two SNPs were conducted using TaqMan assays (Applied Biosystems, Inc. [ABI], Foster City, CA, USA) following the manufacturer’s instructions. For analysis, subjects were grouped into e4 carrier vs non-e4 carriers.

### 2.5. Statistical Analysis

Baseline differences in demographic and clinical variables were tested using analysis of variance for continuous variables and chi-square for categorical variables. For both cross-sectional and longitudinal analysis, the base model (Model 1) was adjusted for age, sex, and study site (Greve in Chianti or Bagno a Ripoli). A second model adjusted for important factors associated with cognitive function, including number of chronic diseases, total energy, years of education, physical activity, smoking, BMI, and apo e4 status (Model 2). In the third model, to investigate the mediating role of inflammation in the relationship between MedDiet and cognition, inflammatory markers for CRP and IL-6 were included (Model 3). Finally, to examine the effect of potential dietary mediators on the relationship, circulating levels of omega-3 fatty acids, omega-6 fatty acids, beta-carotene, and alpha tocopherol were included in the model as covariates (Model 4). If inclusion of the potential mediator variables caused a change in the size of the regression coefficient expressing the independent relationship between MedDiet and cognition by over 30%, that variable was considered as a mediator. The cross-sectional association between MDS and cognitive status (cognitive normal, cognitive impairment, and dementia) was assessed using multinomial logistic regression, treating MDS categories as categorical (low, medium, and high adherence). Odds ratios for the risk of cognitive impairment and dementia were calculated for the MDS categories using the low adherence group as a reference. Correlation between MDS with MMSE in those without cognitive impairment was assessed using multiple linear regression.

The longitudinal analysis of cognition was analyzed as a dichotomous even (time to decline of 5 units in MMSE) and as continuous trajectories of the 18-year follow up period. To estimate the risk of cognitive decline as a dichotomous variable, cox proportional hazard models were performed to analyze time to first instance of 5 or more units of decline in MMSE using coxph function in R. The main predictor variable MDS was assessed as a continuous or categorical variable using low adherence as the reference group. For MMSE trajectory analysis, repeated measures of MMSE over the 5 visits were analyzed using linear mixed model, with MDS as a continuous or categorical variable. To assess the differences in trajectories of MDS, we examined the interaction between time (years of follow up) by MDS. Significant interaction would suggest that there were differences in the slope of MMSE trajectories by MDS categories. For the continuous analysis, the beta for the interaction would signify the unit change in the slope of MMSE decline with an increase in 1 MDS score. For the categorical analysis, the betas would reflect the average difference in slope of MMSE for those in the medium MedDiet adherence or the high MedDiet adherence group relative to the low adherence group. In addition to assessment of MDS, we also examined the association of each of the nine food group constituents of the MDS individually. In these analysis, the other 8 food groups were included in the model as covariates. All analyses were conducted using R version 3.4.2.

## 3. Results

The characteristics of the InCHIANTI study population overall and by MDS groups are presented in Table 1.

At the baseline visit, those with higher MDS were younger, reported more years of education, had fewer chronic diseases, higher MMSE score, higher activity levels, higher plasma beta carotene, and higher alpha tocopherol. Overall, 56% of the cohort were women, with highest percentage of women in the medium adherence group, while lower percentages were found in the highest adherence group. No differences were found for BMI and prevalence of smoking. Except for meat consumption, there were significant differences in the mean intake of eight food groups contributing to the construction of the MDS. As expected, those in higher MDS adherence groups consumed greater amounts of vegetables, legumes, fruits and nuts, cereal, fish, alcohol, monounsaturated fatty acid to saturated fatty acid (MUFA:SFA) ratio, and lower dairy products.

Of the 1139 subjects at baseline, 7% (*N* = 79) had dementia, and 20% (*N* = 228) had cognitive impairment. There were no significant associations between MDS adherence and prevalence of dementia (Odds ratio(OR)medium = 0.65, 95% CI: 0.37–1.12), ORhigh = 1.21, 95% CI: 0.75–1.93) or cognitive impairment (ORmedium = 1.19, 95% CI: 0.90–1.57), ORhigh = 0.83, 95% CI: 0.64–1.07). Among those who were cognitively normal (*N* = 832), there was no association between MDS adherence and MMSE score at baseline (ßmedium = −0.076 ± 0.17 *p* = 0.65; ßhigh = −0.15 ± 0.185, *p* = 0.419).

To examine the association between MDS and longitudinal change in cognitive function, risk of cognitive decline was assessed as time to first follow-up visit where MMSE declined by 5 or more units. Of the 832 subjects that were cognitive normal at baseline, 28.5% or 237 subjects experienced cognitive decline defined as 5 or more unit decrease in MMSE in subsequent visits. There was significant association between MDS and cognitive decline over the 18.2 year follow up period (Table 2).

From the analysis of MDS as a continuous variable, there was significant reduction in risk of cognitive decline, where with every increase in MDS adherence score, there was a 13% reduction in the risk of experiencing cognitive decline (Hazard Ratio (HR) = 0.87, 95% CI: 0.80–0.97). This association remained significant after adjustment for demographic and clinical confounders (HR = 0.89, 95% CI: 0.81–0.97), inflammation (HR = 0.89, 95% CI: 0.81–0.97), and circulating nutrients (HR = 0.88, 95% CI: 0.80–0.97). In the categorical analysis of MDS, compared to those in the group with the lowest adherence to MedDiet, subjects in the high adherence group were less likely to be decliners (HR = 0.60, 95% CI: 0.42–0.85). This association remained significant in the fully adjusted model (HR = 0.59, 95% CI: 0.39–0.88). Those in the medium adherence group also had lower risk of decline (HR = 0.73, 95% CI: 0.54–0.97), however, the association was no longer significant after adjustment for covariates (HR = 0.75, 95% CI: 0.53–1.05). Examining the nine food groups individually, total vegetables (HR = 0.61, 95%CI: 0.47–0.80), and fish (HR = 0.72, 95%CI: 0.56–0.94), had independent protective effects on cognitive decline, however the association was not significant in the fully adjusted model (Table 2).

To further examine the association between MDS and MMSE decline in more detail, we modeled the trajectories of MMSE using a linear mixed model. The mean follow-up period was 10 years (maximum of 18.2 years), and the average decline in MMSE was a decline of 0.4 units per year. The trajectory of MMSE decline was significantly different by MDS, where every unit increase in MDS was associated with 0.085 unit attenuation in MMSE decline, and the protective effect remained after adjustment for covariates (b = 0.08, se = 0.02, *p* < 0.001; Table 3). There were significant differences between the high adherence group compared to low adherence group, where annual decline in MMSE was attenuated on average by 0.34 units (p<0.001; Table 3, Figure 1). After adjustment for multiple covariates, the attenuation remained significant (b = 0.35, se = 0.08, *p* < 0.001). The medium adherent group also tended to have attenuated decline, with average annual decline attenuation of 0.13 units, however the association was not significant (*p* = 0.055) but became significant after adjustment for covariates (b = 0.16, se = 0.07, *p* = 0.028). When analyzing the food groups individually, higher consumption of vegetables, fish, legumes, MUFA:SFA ratio, and fruit and nuts intake was associated with attenuated decline in MMSE by 0.31, 0.18, 0.12, 0.12, and 0.13 units per year, respectively, and lower consumption of meat was associated with accelerated decline in MMSE of 0.13 units per year (Table 3). This association remained significant after adjustment for multiple confounders (Table 3).

## 4. Discussion

In this longitudinal study, higher adherence to MedDiet was protective of cognitive decline over an 18.2-year follow-up period. While we found no significant association between MedDiet and prevalent dementia and cognitive impairment at baseline, there were significant dose-dependent prospective effects of MDS on longitudinal change in cognitive function. While both the medium and high adherence group showed protective effects on cognitive decline, the effects were stronger in those in the highest adherence group. This relationship was independent of some key risk factors including baseline age, apo e4 carrier status, years of education, inflammation, physical activity, total energy intake, circulating levels of and potential mediators for CRP, IL-6, plasma beta-carotene, omega-6, omega-3, and vitamin E. In the InCHIANTI study, the protective effect of MedDiet was strongest for vegetable consumption, followed by fish and legumes.

Our findings are consistent with a recent meta-analysis that examined the association between MedDiet and risk of developing cognitive disorders in prospective cohort studies [11]. The analysis included results from prospective cohorts based in the United States [25,26,27,28], France [8], Australia [29], and Sweden [30]. The cohorts represented varying lengths of follow-up periods, ranging from 2.2 to 9.1 years. In the pooled estimates, those in the high adherence group were 21% less likely to develop dementia, Alzheimer’s disease, and mild cognitive impairment [11]. No significant association was observed for those in the medium adherence group. When MDS was analyzed as a continuous variable, there was a 6% reduction in the risk of cognitive disorders. These results are consistent with the results of the current study, where we observed protective effects of MedDiet on cognitive decline, particularly on those in the high adherence group. We also observed an MDS dosage effect of 16% reduction per unit increase in MDS. We also report differences in trajectories of MMSE by MedDiet adherence group, where higher adherence to MedDiet was associated with slower decline. This is consistent with one study in 1410 French men and women over 65 years of age, where higher adherence to MedDiet was associated with slower MMSE decline over a 4-year follow up period [8]. Our results are consistent with the finding reported above, and further suggest that the beneficial effects of MedDiet on MMSE decline is long-lasting and can be observed up to 18 years after the dietary assessment.

The protective effect of MedDiet on cognitive health is supported by results from the PREDIMED study, a randomized controlled intervention trial of MedDiet supplements with olive oil or nuts. At the end of an intervention period (median follow up of ~4 years), subjects consuming MedDiet supplemented with olive oil or nuts showed slower decline in composite memory, frontal cognition, and global cognitive score [31,32]. The totality of the evidence from prospective observational studies, including ours, and results from the intervention study, support that MedDiet is an effective dietary strategy to maintain cognitive health.

There are multiple possible mechanisms that mediate the protective association of MedDiet with cognitive decline. MedDiets are enriched in nutrients that have been shown to be neuroprotective, including carotenoids, vitamin E, polyphenols, and polyunsaturated fatty acids [16,33,34,35,36]. There are many hypothesized mechanisms for neuroprotection by these nutrients, including prevention through their antioxidant properties and reduced inflammation. To this end, higher adherence to MedDiet has been associated with increased total antioxidant capacity and reduced concentrations of inflammatory markers, such as CRP [37]. In the current study, the associations between MedDiet and cognition remained after adjustment for baseline levels of carotenoids, vitamin E, CRP, and omega-3 fatty acids, suggesting that there are other mechanisms that are mediating the association. Alternatively, MedDiet may be influencing the trajectories of these confounders, thus, future studies may consider time-varying covariates into the model.

In addition to the direct effect of MedDiet on neurological health, the positive effect on cognition may also be through the promotion of overall health. Observational studies have shown that adherence to MedDiet has a positive effect on several diseases and disease risk factors, such as diabetes, obesity, and heart disease [38,39,40]. The PREDIMED randomized intervention trial showed that compared to a control reduced fat diet, MedDiet supplemented with olive oil or nuts had a reduction in the hazard of cardiovascular disease events by 31% and 28%, respectively [41]. There are many shared risk factors for cognitive decline with these age-related diseases, thus it is plausible for MedDiet to have a protective effect on age-related diseases, including cognitive function, through shared mechanisms, such as inflammation.

There are many strengths to our study. First, the InCHIANTI study is a well characterized study that is representative of the general Italian population. Thus, we believe that the results are generalizable to populations over the age of 65. Second, we had extensive follow-up observations of up to 18 years that allowed us to assess the long-term effect of diet on cognitive function. Third, due to the richness of the clinical data, we were able to account for many of the major factors that may confound the relationship between MedDiet and cognitive decline, including genetics, as well as demographic and clinical characteristics. There are also several limitations to our study. This study is an observational study, thus the effects that we reported may be due to confounders that were not measured in our study. The MedDiet score was estimated using a dietary assessment instrument, the FFQ, that is known to introduce some measurement error [42]. While the MMSE is a widely accepted method to assess cognitive function, there are some limitations, including influences by non-cognitive domains (such as education), and also may not be a sensitive test for detecting early signs of dementia [43]. Future studies should consider other cognitive tests to confirm the findings of this study. In the current analysis, we did not consider MedDiet changes over time, thus future studies should consider analyzing longitudinal changes in diet. Finally, there were no clinical assessments of dementia in the follow-up visits, thus we were not able to analyze the link between adherence to MedDiet and incidences of dementia in this cohort.

## 5. Conclusions

In summary, the present study contributes to the growing body of evidence of the protective effects of MedDiet on long-term cognitive health. Dementia and cognitive impairment are a growing public health problem that require effective preventative strategies that can be implemented with ease to individuals. Our study supports that dietary recommendations towards a MedDiet could be instrumental in slowing the rates of cognitive decline.

## Figures and Tables

**Figure 1 nutrients-10-02007-f001:**
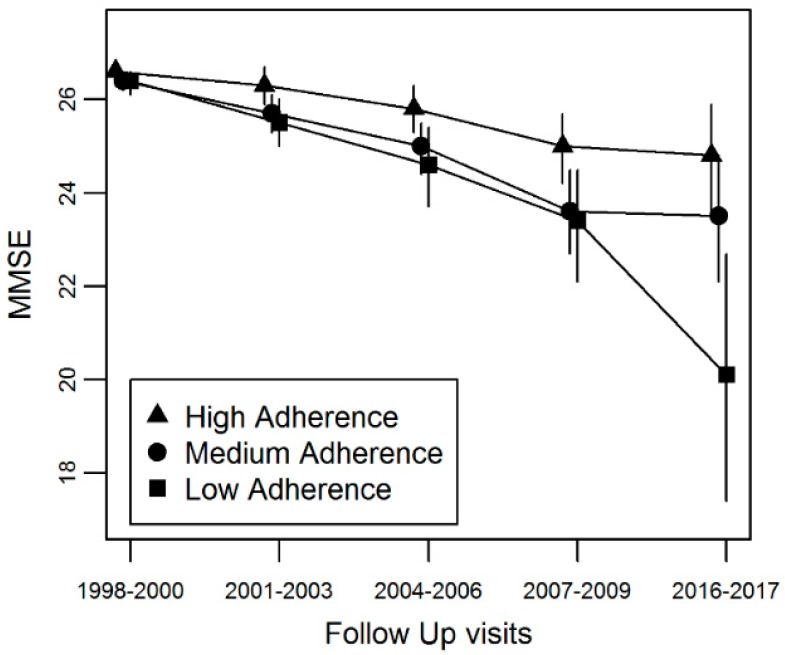
Trajectories of MMSE by Mediterranean Diet Score groups. The graphs display the mean MMSE at baseline and four follow up visits. The high adherence group (triangle) has the least decline in MMSE over time, followed by the medium adherence group (circle) and the low adherence group (square).

**Table 1 nutrients-10-02007-t001:** Characteristics of the InChianti study participants aged 65 y and older at baseline.

	All	Low MDS (≤3)	Med MDS (4–5)	High MDS (≥6)	*p*
*n*	1139	356	487	296	
Age, years	75.4 ± 7.6	77.7 ± 8.2	75.2 ± 7.4	72.9 ± 6.1	<0.001
Female, *N* (%)	644 (56.5)	205 (57.6)	299 (61.4)	140 (47.3)	<0.001
Years of Education	5.3 ± 3.3	5.1 ± 3.6	5.1 ± 3.1	5.8 ± 3.4	0.020
Chronic Diseases	1.4 ± 1.3	1.6 ± 1.5	1.4 ± 1.2	1.3 ± 1.2	0.004
Mini mental state examination score	24.2 ± 5.3	22.9 ± 6.6	24.5 ± 4.7	25.3 ± 3.8	<0.001
Body Mass Index, kg/m^2^	27.5 ± 4.1	27.1 ± 4.3	27.5 ± 4.3	27.8 ± 3.7	0.154
C-reactive protein (ug/mL)	5.5 ± 9.6	6.6 ± 12.0	4.6 ± 5.7	5.5 ± 11.2	0.331
Interleukin-6 (pg/mL)	2.3 ± 4.3	2.5 ± 3.0	2.1 ± 2.1	2.5 ± 7.2	0.969
Plasma Omega 3 (% total fatty acid)	1.9 ± 0.6	1.9 ± 0.6	2.0 ± 0.6	2.0 ± 0.7	0.072
Plasma Omega 6 (% total fatty acids)	29.7 ± 4.4	29.7 ± 4.4	29.8 ± 4.5	29.3 ± 4.2	0.539
Beta-carotene (μmol/L)	0.42 ± 0.3	0.36 ± 0.2	0.44 ± 0.3	0.44 ± 0.3	0.001
Alpha tocopherol (μmol/L)	33.6 ± 7.6	32.3 ± 7.8	34.0 ± 7.6	34.3 ± 7.1	0.005
Smoking, *n* (%)					0.554
Non-Smoker	677 (59.4)	216 (60.7)	297 (61)	164 (55.4)	
Former Smoker	302 (26.5)	89 (25)	125 (25.7)	88 (29.7)	
Smoker	160 (14)	51 (14.3)	65 (13.3)	44 (14.9)	
Physical Activity, *n* (%)					<0.001
Low	269 (23.7)	133 (37.5)	96 (19.9)	40 (13.5)	
Medium	479 (42.2)	138 (38.9)	211 (43.7)	130 (43.9)	
High	386 (34)	84 (23.7)	176 (36.4)	126 (42.6)	
Apolipoprotein E4 carriers, *n* (%)	119 (15.2)	30 (14.4)	56 (16.5)	33 (14.2)	0.692
Dietary Components					
Total Energy, kcal/day	1892 ± 563	1734 ± 558	1887 ± 551	2092 ± 526	<0.001
Vegetables, g/day	152.3 ± 90.4	100.2 ± 63.4	154.7 ± 84.1	210.9 ± 90.9	<0.001
Legumes, g/day	16.5 ± 10.5	11.5 ± 8.0	17.3 ± 10.6	21.4 ± 10.5	<0.001
Fruits and nuts, g/day	284.1 ± 135.7	228.6 ± 122.3	287.8 ± 128.1	344.8 ± 136.1	<0.001
Cereal, g/day	237.3 ± 98.4	215.1 ± 96.0	236.4 ± 97.4	265.4 ± 96.2	<0.001
Fish, g/day	23.2 ± 17.5	16.5 ± 12.7	23.4 ± 18.0	30.7 ± 18.4	<0.001
Monounsaturated:saturated lipid	1.5 ± 0.4	1.3 ± 0.3	1.5 ± 0.3	1.8 ± 0.4	<0.001
Meat, g/day	104.8 ± 43.8	101.4 ± 41.5	106.6 ± 44.4	106.0 ± 45.4	0.208
Dairy, g/day	170.3 ± 141.7	205.0 ± 142.4	167.0 ± 153.6	134.0 ± 106.5	<0.001
Alcohol, g/day	13.8 ± 19.6	11.5 ± 19.7	14.1 ± 21.0	16.1 ± 16.7	0.011

MDS: Mediterranean Diet Score. Values are mean ± SD for continuous characteristics and *n* (%) for categorical characteristics.

**Table 2 nutrients-10-02007-t002:** Longitudinal associations between Mediterranean Diet Score and its components at baseline and cognitive decline after a follow-up of 18 y among InCHIANTI participants 65 y and older.

		Model 1			Model 2			Model 3			Model 4		
		HR ^+^	95% CI	*p*	HR ^+^	95% CI	*p*	HR ^+^	95% CI	*p*	HR ^+^	95% CI	*p*
MDS *	Continuous	0.87	(0.8, 0.95)	0.001	0.89	(0.81, 0.97)	0.007	0.89	(0.81, 0.97)	0.007	0.88	(0.80, 0.97)	0.009
	Medium vs Low Adherence **	0.73	(0.54, 0.97)	0.033	0.76	(0.55, 1.05)	0.094	0.74	(0.54, 1.02)	0.068	0.75	(0.53, 1.05)	0.089
	High vs Low Adherence **	0.60	(0.42, 0.85)	0.005	0.62	(0.42, 0.9)	0.013	0.62	(0.43, 0.91)	0.015	0.59	(0.39, 0.88)	0.011
Optimal consumption of individual food groups ^+^	Vegetables	0.61	(0.47, 0.8)	0	0.73	(0.52, 1.03)	0.073	0.74	(0.53, 1.04)	0.083	0.74	(0.51, 1.07)	0.106
	Legume	0.80	(0.61, 1.03)	0.083	0.78	(0.59, 1.04)	0.094	0.77	(0.57, 1.02)	0.070	0.74	(0.54, 1.00)	0.048
	Fish	0.72	(0.56, 0.94)	0.015	0.81	(0.61, 1.07)	0.13	0.80	(0.60, 1.06)	0.115	0.86	(0.64, 1.16)	0.325
	Fruits and nuts	0.82	(0.64, 1.06)	0.133	0.91	(0.69, 1.21)	0.515	0.93	(0.70, 1.23)	0.597	0.93	(0.69, 1.26)	0.636
	Cereal	0.96	(0.74, 1.24)	0.747	0.94	(0.66, 1.35)	0.738	0.94	(0.66, 1.35)	0.748	0.90	(0.62, 1.33)	0.608
	MUFA:SFA***	0.88	(0.68, 1.14)	0.331	0.97	(0.71, 1.33)	0.853	0.95	(0.69, 1.31)	0.767	0.89	(0.64, 1.25)	0.510
	Dairy	1.13	(0.88, 1.46)	0.346	1.13	(0.85, 1.49)	0.41	1.14	(0.86, 1.51)	0.380	1.11	(0.82, 1.50)	0.492
	Meat	1.12	(0.86, 1.46)	0.389	1.14	(0.85, 1.55)	0.385	1.13	(0.84, 1.53)	0.417	1.12	(0.81, 1.54)	0.499
	Alcohol	0.82	(0.63, 1.06)	0.133	0.81	(0.61, 1.08)	0.159	0.81	(0.61, 1.08)	0.156	0.85	(0.63, 1.15)	0.286

* MDS: Mediterranean Diet Score; ** Low adherence group MDS ≤ 3, medium adherence has score between 4–5, and high adherence is score ≥ 6. Cognitive decline defined as a decline in 5units of MMSE at any of the follow up visit; Model 1 adjusted for baseline age, sex, study site; Model 2 adjusted for Model 1 plus chronic diseases, years of education, total energy intake, physical activity, BMI, ApoE4 carrier status; Model 3 adjusted for Model 2 plus CRP, IL-6; Model 4 adjusted for Model 3 plus plasma omega-3, plasma omega-6, plasma beta-carotene, and plasma alpha-tocopherol; ^+^ Optimal food groups are defined as above median levels for vegetables, legumes, fruits and nuts, cereal, fish, and MUFA:SFA, and below median levels for meat and dairy foods. Alcohol consumption of 5–25 g/day and 10–50 g/day for women and men was considered optimal; ^+^hazard ratios (HR) and 95% confidence interval (CI) for cognitive decline for continuous MDS is the odds of cognitive decline per 1 unit of MDS score; *** Monounsaturated fatty acid to saturated fatty acid (MUFA:SFA).

**Table 3 nutrients-10-02007-t003:** Association of Mediterranean Diet Score with MMSE longitudinal trajectories.

	Base Model 1			Base Mode1 2			Base Model 3			Base Model 4		
	Beta **	SE	*p*	Beta **	SE	*p*	Beta **	SE	*p*	Beta **	SE	*p*
Continuous	0.085	0.017	<0.001	0.076	0.017	<0.001	0.077	0.017	<0.001	0.081	0.018	<0.001
Medium vs Low Adherence *	0.131	0.068	0.055	0.124	0.069	0.071	0.125	0.069	0.070	0.158	0.072	0.028
High vs Low Adherence *	0.342	0.073	<0.001	0.313	0.073	<0.001	0.316	0.074	<0.001	0.346	0.077	<0.001
Vegetables ^+^	0.314	0.054	<0.001	0.278	0.054	<0.001	0.279	0.055	<0.001	0.285	0.057	<0.001
Fish ^+^	0.179	0.055	0.001	0.148	0.055	0.007	0.148	0.056	0.008	0.118	0.058	0.044
Legume ^+^	0.121	0.055	0.029	0.118	0.055	0.032	0.122	0.056	0.029	0.125	0.058	0.032
Fruits and nuts ^+^	0.120	0.056	0.030	0.114	0.056	0.040	0.116	0.056	0.040	0.140	0.059	0.017
Cereal ^+^	0.099	0.055	0.073	0.067	0.055	0.226	0.066	0.056	0.236	0.081	0.058	0.163
MUFA:SFA ^+^	0.125	0.055	0.024	0.114	0.055	0.039	0.114	0.056	0.041	0.145	0.058	0.012
Dairy ^+^	0.012	0.055	0.832	0.028	0.055	0.614	0.029	0.056	0.609	0.023	0.058	0.695
Meat ^+^	−0.132	0.055	0.018	−0.127	0.055	0.021	−0.127	0.056	0.023	−0.139	0.058	0.017
Alcohol ^+^	0.056	0.056	0.317	0.065	0.056	0.247	0.064	0.057	0.258	0.039	0.059	0.506

Model 1 adjusted for baseline age, sex, study site. Model 2 adjusted for Model 1 plus chronic diseases, years of education, total energy intake, physical activity, BMI, ApoE4. Model 3 adjusted for Model 2 plus CRP, IL-6. Model 4 adjusted for Model 3 plasma omega-3, plasma omega-6, plasma beta-carotene, and plasma alpha-tocopherol; individual food group analysis is adjusted for all other food groups; * Low adherence group MDS < 3, medium adherence has score between 4–5, and high adherence is score > 6. ^+^ Optimal food groups are defined as above median levels for vegetables, legumes, fruits and nuts, cereals, fish, and MUFA:SFA, and below median levels for meat and dairy foods. Alcohol consumption of 5–25g/day and 10–50g/day for women and men was considered optimal; ** The beta estimate represents the difference in slope from the average decline of MMSE of 0.4 units per year with positive values indicating attenuation and negative values reflecting amplification of decline for the modeled group. SE: standard error.

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
