# Peer review of "Adherence to a Mediterranean Diet Protects from Cognitive Decline in the Invecchiare in Chianti Study of Aging"

_nutrients, 2018, doi:10.3390/nu10122007_

Reviewer 1 Report

The manuscript is well-structured with straightforward exposures and analysis, adding to the evidence base for the role of Mediterranean-type diet and cognitive health. A few comments below:

1.      Lines 81-84: the MDS scoring, is it equal to or above the median or just above?

2.      Lines 93-95: not clear what authors mean. When/ in what context did participants report ADLs/IADLs difficulties? Please clarify.

3.      Please refrain from using the term ‘demented’, as according to recent guidelines it can be considered disrespectful and stigmatising by people living with dementia and their families.

4.      Lines 114-115: Why did authors choose the 5-point decrease as a cutoff for decline? Please add explanation and reference to support that decision.

5.      Lines 149-151: Please clarify how mediation was tested, as below when explaining the interpretation of significant interaction. Where is your mediation results reported?

6.      Table 1: add APOE status;  add footnote that characteristics are presented as mean ± SD where appropriate;

Table 2: line 211 replace ‘MeDiet’ with ‘MDS’?

7.      Lines 236-239: It appears that authors have misinterpreted the results for meat consumption. Please correct description.

8.      Line 267: Is it cognitive impairment, not decline at baseline?

9.      Lines 286-291: paragraph needs editing or re-wording as it is hard to read and comprehend.

10.  The authors use the PREDIMED trial to discuss similarities with existing literature. The trial was recently heavily criticised for problems in the randomisation process, and relevant papers with cardiovascular disease as primary outcomes have been retracted and re-analysed. Therefore, I would urge the authors to double-check they are citing the re-analysed studies and whether results reporting cognitive outcomes have also been affected.

11.  Authors should consider adding limitations of the MMSE in their discussion.

12.  Although generally well-written, there are numerous typos and grammatical errors throughout including the name of the study (line 174) and the inconsistent use of MeDiet/MedDiet. Please proofread and edit.

Author Response

1.      Lines 81-84: the MDS scoring, is it equal to or above the median or just above?

>Thank you for asking this questions. It is actually above or equal to. We have changed the text to reflect this.

2.      Lines 93-95: not clear what authors mean. When/ in what context did participants report ADLs/IADLs difficulties? Please clarify.

>As part of the assessment of ability to perform ADL/IADL, if they answered that they had difficulties in performing any of the tasks, they are asked if this is was due to cognitive impairment. We agree with the reviewer that this sentence is confusing so we have reworded this sentence.

3.      Please refrain from using the term ‘demented’, as according to recent guidelines it can be considered disrespectful and stigmatising by people living with dementia and their families.

>Thank you for pointing this out. We have removed the word “demented” from the text.

4.      Lines 114-115: Why did authors choose the 5-point decrease as a cutoff for decline? Please add explanation and reference to support that decision.

>We have included the appropriate reference for the decision to use 5 point decreased scale to define cognitive decline.

5.      Lines 149-151: Please clarify how mediation was tested, as below when explaining the interpretation of significant interaction. Where is your mediation results reported?

> We tested for mediation, not moderation. To test for this, we included the potential mediator variable in the model, and if this resulted in the change in the magnitude of the regression coefficient between MedDiet and cognition over 30%, that variables was considered as a mediator. We clarified this point in the text.

6.      Table 1: add APOE status;  add footnote that characteristics are presented as mean ± SD where appropriate;

>We included the ApoE4 carrier counts and the footnote suggested

Table 2: line 211 replace ‘MeDiet’ with ‘MDS’?

>This change was made.

7.      Lines 236-239: It appears that authors have misinterpreted the results for meat consumption. Please correct description.

>In creating the MDS score, meat is considered one of the detrimental groups, thus those who have a score of 1 are those who consume less meats. The beta from the interaction term is negative suggesting that consuming less meat is associated with greater decline in MMSE, conversely, consuming more meat would have lesser decline. This would suggest that meat consumption is not detrimental for cognitive decline in our study population. We agree with the reviewers that this is confusing, and we have rewritten the results.

8.      Line 267: Is it cognitive impairment, not decline at baseline?

>Thank you for pointing out this typo. We have changed it accordingly.

9.      Lines 286-291: paragraph needs editing or re-wording as it is hard to read and comprehend.

>Thank you for pointing out the confusion. We have reworded this paragraph and believe it is clearer.

10.  The authors use the PREDIMED trial to discuss similarities with existing literature. The trial was recently heavily criticised for problems in the randomisation process, and relevant papers with cardiovascular disease as primary outcomes have been retracted and re-analysed. Therefore, I would urge the authors to double-check they are citing the re-analysed studies and whether results reporting cognitive outcomes have also been affected.

>We thank the reviewer for bringing attention to this important point. The discussion of CVD outcomes references the reanalyzed version of the article. Further we have removed the discussion based on the first paper of cognitive outcomes from the PREDIMED and included the reanalyzed results published in JAMA internal medicine (December 2018) and include the appropriate reference.

11.  Authors should consider adding limitations of the MMSE in their discussion.

>A discussion of the limitation of MMSE was included in the discussion. (line 338-341)

12.  Although generally well-written, there are numerous typos and grammatical errors throughout including the name of the study (line 174) and the inconsistent use of MeDiet/MedDiet. Please proofread and edit.

>We have changed the errors and inconsistencies in the text. Thank you for pointing this out.

Reviewer 2 Report

I think your manuscript is well described and written. 

a) Please mention the approval and payment procedure for using MMSE?

b) How you validate the physical activity interview?

c) What are the reason behind not taking ADL as a co-variate in Model 2?

Author Response

a) Please mention the approval and payment procedure for using MMSE?

>The InCHIANTI study used an Italian version of MMSE that was solely used for research purposes and no royalties were paid. We have included that an Italian version of MMSE was utilized.

b) How you validate the physical activity interview?

>Thank you for pointing this out. The questionnaire used in this analysis has not been validated, but is derived from NHANES questionnaire that has been validated, the reference for this questionnaire is included in the text (reference 20). Physical activity has been associated with important functional outcomes thus we feel is adequate to use as a covariate.

c) What are the reason behind not taking ADL as a co-variate in Model 2?

>For most of the analysis of trajectories, we use subjects who do not have any cognitive impairment or dementia at baseline. This means that they have reported no difficulties in ADL or IADL. For this reason, we did not include these variables in the model.